# Exploring Four *Atraphaxis* Species: Traditional Medicinal Uses, Phytochemistry, and Pharmacological Activities

**DOI:** 10.3390/molecules29040910

**Published:** 2024-02-19

**Authors:** Alima Abilkassymova, Aknur Turgumbayeva, Lazzat Sarsenova, Kuanysh Tastambek, Nazym Altynbay, Gulnar Ziyaeva, Ravil Blatov, Gulmira Altynbayeva, Kuralay Bekesheva, Gulzhamal Abdieva, Perizat Ualieva, Zhanserik Shynykul, Assem Kalykova

**Affiliations:** 1Higher School of Medicine, Al-Farabi Kazakh National University, Almaty 050040, Kazakhstan; sarsenova.lazzat@med-kaznu.com (L.S.); shynykul.zhanserik@med-kaznu.com (Z.S.); a.kalykova@gmail.com (A.K.); 2School of Life Sciences, University of Westminster, 115 New Cavendish Street, London W1W 6UW, UK; 3Institute of Ecology, Khoja Akhmet Yassawi International Kazakh-Turkish University, Turkistan 161200, Kazakhstan; tastambeku@gmail.com; 4Institute of Ecological Problems, Al-Farabi Kazakh National University, Al-Farabi Ave. 71, Almaty 050040, Kazakhstan; altynbaynazym@gmail.com; 5Department of Biology, Taraz Regional University Named after M.Kh.Dulaty, Taraz 080000, Kazakhstan; ziyayevagulnar@gmail.com; 6Department of Pharmacy, Kazakh-Russian Medical University, Almaty 050000, Kazakhstan; rblatov02@gmail.com; 7School of Pharmacy, JSC “S.D. Asfendiyarov Kazakh National Medical University”, Almaty 050000, Kazakhstan; altynbayeva.g85@gmail.com; 8Neonatology and Neonatal Surgery Department, JSC “Scientific Center of Pediatrics and Pediatric Surgery”, Almaty 050060, Kazakhstan; 9JSC “Scientific Centre for Anti-Infectious Drugs”, Almaty 010000, Kazakhstan; kuralayaryn02@gmail.com; 10Department of Biotechnology, Faculty of Biology and Biotechnology, Al-Farabi Kazakh National University, Al-Farabi 71, Almaty 050040, Kazakhstan; abdievagzh@gmail.com (G.A.); ualievaps@gmail.com (P.U.)

**Keywords:** *Atraphaxis*, Polygonaceae, *Atraphaxis laetevirens*, *Atraphaxis frutescens*, *Atraphaxis spinosa* L., *Atraphaxis pyrifolia* medicinal plants, phytochemistry, ethnopharmacology

## Abstract

*Atraphaxis* is a genus of flowering plants in the family Polygonaceae, with approximately 60 species. Species of *Atraphaxis* are much-branched woody plants, forming shrubs or shrubby tufts, primarily inhabiting arid zones across the temperate steppe and desert regions of Central Asia, America, and Australia. *Atraphaxis* species have been used by diverse groups of people all over the world for the treatment of various diseases. However, their biologically active compounds with therapeutic properties have not been investigated well. Studying the biologically active components of *Atraphaxis laetevirens*, *Atraphaxis frutescens*, *Atraphaxis spinosa* L., and *Atraphaxis pyrifolia* is crucial for several reasons. Firstly, it can unveil the therapeutic potential of these plants, aiding in the development of novel medicines or natural remedies for various health conditions. Understanding their bioactive compounds enables scientists to explore their pharmacological properties, potentially leading to the discovery of new drugs or treatments. Additionally, investigating these components contributes to preserving traditional knowledge and validating the historical uses of these plants in ethnomedicine, thus supporting their conservation and sustainable utilization. These herbs have been used as an anti-inflammatory and hypertension remedies since the dawn of time. Moreover, they have been used to treat a variety of gastrointestinal disorders and problems related to skin in traditional Kazakh medicine. Hence, the genus *Atraphaxis* can be considered as a potential medicinal plant source that is very rich in biologically active compounds that may exhibit great pharmacological properties, such as antioxidant, antibacterial, antiulcer, hypoglycemic, wound healing, neuroprotective, antidiabetic, and so on. This study aims to provide a collection of publications on the species of *Atraphaxis*, along with a critical review of the literature data. This review will constitute support for further investigations on the pharmacological activity of these medicinal plant species.

## 1. Introduction

Phytochemical screening of plant extracts is of paramount importance due to the myriad benefits it offers in various sectors [1]. These screenings are crucial for several reasons. Firstly, phytochemical screening aids in the discovery and identification of bioactive compounds in plants, which have the potential to be harnessed for medicinal purposes. Many pharmaceutical drugs have their origins in plant-based compounds, making this screening vital for drug development [2,3,4]. Secondly, it plays a significant role in the field of nutrition and health. The identification of phytochemicals, such as flavonoids, carotenoids, and polyphenols, in plant extracts helps in understanding their health-promoting properties. These compounds have antioxidant, anti-inflammatory, and anti-cancer effects, making them valuable for dietary and nutritional purposes [5,6,7]. Furthermore, phytochemical screening contributes to the field of traditional medicine and alternative therapies. Traditional herbal remedies often rely on the use of plant extracts, and screening helps validate their efficacy and safety [8]. Lastly, the industrial sector benefits from phytochemical screening, as certain compounds found in plants have applications in cosmetics, agriculture, and food processing [9]. Overall, phytochemical screening of plant extracts is instrumental in unlocking the potential of nature’s pharmacy, with far-reaching implications for medicine, nutrition, traditional healing, and industry. It continues to be a key driver in scientific exploration and innovation.

The genus *Atraphaxis* belongs to the family Polygonaceae and comprises many species and subspecies [10,11,12,13]. *Atraphaxis* is highly drought-tolerant and inhabits areas along the foothills of mountains and edges of deserts [14,15]. Frequently, *Atraphaxis* species constitute a significant source of feed for livestock [16,17]. Some have been exploited as a commercial supply of lye or potash, while others have served as important fuel sources in *Atraphaxis*. Species of *Atraphaxis* are much-branched woody plants, forming shrubs or shrubby tufts. The current year’s branchlets are herbaceous and bear the leaves and flowers [18,19]. The leaves are simple and alternate, with very short stalks. The ochreas are membranous and usually two-veined, more-or-less joined at the base. The inflorescence is made up of several bundles (fascicles) of one to three flowers. The flowers have persistent tepals, either arranged in a narrow tube with unequal lobes or bell-shaped with equal segments. The fruits are wingless achenes [20,21,22,23]. These plants’ most notable characteristic is their resistance to saline environments and environmental harsh conditions [24,25,26].

In 1753, Carl Linnaeus established the genus *Atraphaxis* [27]. Like many other genera within the Polygonaceae family, the distinctions between these genera have often been unclear, leading to the reassignment of some or all species into different genera [28]. According to some molecular phylogenetic studies, *Atraphaxis* forms a distinct clade, and this genus is categorized within the Polygoneae tribe of the Polygonoideae subfamily, being part of the so-called “DAP clade” within the tribe, which is most closely related to the genera *Duma* and *Polygonum* [29]. The genus *Atraphaxis*, which falls under the Polygonaceae family, consists of over 60 species found across a vast geographical range, spanning from southeastern Europe and northeastern Africa to East Siberia, China, and Mongolia [30,31,32]. Its primary hubs of taxonomic diversity are located in Southwest Asia and Central Asia. In terms of morphological characteristics, *Atraphaxis* has been categorized within the tribes Polygoneae, Rumiceae, Atraphaxideae, or Calligoneae of the Polygonaceae family [33].

*Atraphaxis* species have a history of traditional medicinal use in various regions, particularly in Asia and the Middle East. These plants have been employed for their potential therapeutic properties in traditional medicine systems [34]. In some traditional systems of medicine, *Atraphaxis* species have been used to treat gastrointestinal disorders such as diarrhea, indigestion, and stomachaches. The plants may be consumed as herbal infusions or decoctions to alleviate these symptoms. Some *Atraphaxis* species have been used for their anti-inflammatory properties. They may be applied topically or consumed as herbal remedies to reduce inflammation associated with conditions such as arthritis, joint pain, or skin inflammation in Kazakh medicine. Traditional healers in certain regions have used *Atraphaxis* species to aid in wound healing. The plants may be prepared as poultices or ointments and applied to wounds and cuts to promote the healing process. There is some ethnopharmacological evidence suggesting that *Atraphaxis* species may have potential anti-diabetic properties. Extracts from these plants have been used to manage blood sugar levels in traditional medicine [35]. In some cultures, *Atraphaxis* species have been employed to treat respiratory ailments, such as coughs and asthma. Infusions or extracts from these plants are taken orally or used in a steam inhalation device to relieve respiratory symptoms. *Atraphaxis* species may be used in traditional medicine for their antioxidant properties, which can help combat oxidative stress in the body. Additionally, some traditional practices use these plants to support the immune system. Traditional medicine in certain regions has utilized *Atraphaxis* species to address liver and gallbladder-related issues, including liver detoxification and gallstone management. *Atraphaxis* species are often considered to have astringent properties. This makes them useful in traditional medicine for conditions characterized by excessive bleeding, such as heavy menstrual bleeding. It is important to note that the specific uses of *Atraphaxis* species can vary among cultures and regions, but the efficacy of these traditional remedies may not have been extensively studied or scientifically validated.

Phytochemical investigations of *Atraphaxis* species have unveiled the existence of diverse bioactive compounds, although the precise phytochemical composition can differ among various species and even within the same species [35,36,37,38]. Some of the typical phytochemicals identified in *Atraphaxis* species encompass polyphenols, triterpenoids, alkaloids, essential oils, phenolic acids, saponins, and lignans. *Atraphaxis* species are recognized for containing a range of polyphenolic compounds, such as flavonoids and tannins. These compounds possess antioxidative properties and are believed to play a role in safeguarding the plant against oxidative stress. Triterpenoids, identified in select *Atraphaxis* species, are a class of compounds renowned for their potential anti-inflammatory and cytotoxic characteristics [39]. Certain *Atraphaxis* species have been documented to contain alkaloids, which are nitrogen-containing compounds with a wide array of pharmacological actions, possibly contributing to the plants’ medicinal attributes. Additionally, some *Atraphaxis* species produce essential oils, housing volatile components responsible for the distinctive fragrance of the plant. These essential oils serve various purposes, including traditional medicinal use and flavoring. Phenolic acids, such as caffeic acid and ferulic acid, have been ascertained in *Atraphaxis* species [16]. These compounds offer both antioxidative and anti-inflammatory properties, qualities commonly found in many plant species. Some *Atraphaxis* species contain saponins, glycosides known for their foaming properties, and studied for potential health benefits, including anticancer properties and immune system modulation. Furthermore, *Atraphaxis* species were reported to contain lignans, which are polyphenolic compounds recognized for their antioxidative attributes [40].

The most studied species of the genus *Atraphaxis* are *Atraphaxis spinosa* L. and *Atraphaxis frutescens*. For instance, a chemical analysis of the above-ground components of *A. frutescens*, as conducted by Batsukh O. and colleagues, yielded the separation of five 7-methoxyflavonols featuring pyrogallol B-ring structures. Additionally, they identified a glucoside of fisetinidol and a glycoside of benzyl, along with the presence of 26 previously known compounds, encompassing flavonoids, phenylpropanoid amides, anthraquinone glycosides, lignans, and a derivative of benzyl [35]. According to another study, nine compounds were separated and characterized from an ethereal extract of *A. spinosa* L. var. *sinaica*. These compounds were identified as follows: N-trans-p-coumaroyl-3′,4′-dihydroxyphenylethylamine, N-trans-feruloyl-3′,4′-dihydroxyphenylethylamine, (−)-fisetinidol, (−)-catechin, butin, quercetin, quercetin-3-methyl ether, 5-deoxykaempferol, and β-sitosterol glucoside. Compounds such as N-trans-p-coumaroyl-3′,4′-dihydroxyphenylethylamine and N-trans-feruloyl-3′,4′-dihydroxyphenylethylamine were isolated from natural sources for the first time, and it was demonstrated that they exhibit cytotoxic activity against leukemic P388 cells [36].

The other two species, *Atraphaxis pyrifolia* and *Atraphaxis laetevirens*, have not been fully studied in terms of biologically active compounds. However, a few studies confirmed the presence of some valuable compounds. For instance, chrysophanol, physcion, nepodin, and emodin are natural compounds found in various plants, including *A. laetevirens*. Belonging to a class of organic molecules known as anthraquinones, these compounds have exhibited several pharmacological activities in scientific studies. For instance, chrysophanol and nepodin are well-known for their insecticidal and antibacterial properties [38]. Among the flavonoid glycosides discovered in *A. pyrifolia* were compounds such as 7-methylgossypetin 8-β-D-glucopyranoside 3-O-α-L-rhamnopyranoside and 7-methylgossypetin 8-β-D-glucopyranoside. However, their pharmacological applications have not been investigated.

It is crucial to recognize that the phytochemical composition of *Atraphaxis* species can exhibit significant variations, primarily due to factors such as species diversity, geographical distribution, and environmental conditions. Furthermore, the presence of specific phytochemicals may have noteworthy implications for the traditional medicinal practices of diverse cultures. Researchers continue to investigate the phytochemical profiles of *Atraphaxis* species, aiming to unearth potential pharmacological properties and applications for these compounds. These studies hold immense promise for the development of innovative drugs and natural products, with potential benefits in diverse fields, including medicine, agriculture, and industry. The primary objective of this study was to furnish comprehensive scientific insights into the phytochemistry, traditional uses, and pharmacological attributes of *Atraphaxis* species. This knowledge serves as a foundation for further exploration of the therapeutic potential inherent in these plants. Numerous studies have already demonstrated the ethnobotanical value of various parts of *Atraphaxis* species, including leaves, flowers, seeds, roots, and aerial components, highlighting their multifaceted significance in traditional medicine and beyond. This comprehensive understanding not only aids in the preservation of traditional knowledge but also fuels modern scientific advancements and applications.

## 2. Taxonomic Insights into *Atraphaxis* Species within the Polygonaceae Family

The Polygonaceae family, commonly referred to as the knotweed family or smartweed-buckwheat family in the United States, derives its name from the genus *Polygonum*. This nomenclature was introduced by Antoine Laurent de Jussieu in 1789 in his work “Genera Plantarum”. The name’s origin can be attributed to two possible sources. It may allude to the numerous swollen nodes found on the stems of certain species, drawing from the Greek words “poly” meaning “many” and “gony” meaning “knee” or “joint”. Alternatively, it might have a different root, signifying "many seeds". The Polygonaceae family encompasses approximately 1200 species, distributed among around 48 genera. Notable among these are *Eriogonum* (with 240 species), *Rumex* (200 species), *Coccoloba* (120 species), *Persicaria* (100 species), *Atraphaxis* (60 species), and *Calligonum* (80 species) [17,18,19,20].

Similar to many other genera in the Polygonaceae family, the boundaries between the genera have been ambiguous, leading to the placement of some or all species in other genera. According to molecular phylogenetic analyses, *Atraphaxis* belongs to a separate clade, as shown in Figure 1. The genus is classified in the Polygoneae tribe of the *Polygonoideae* subfamily. It belongs to the so-called “DAP clade” within the tribe, which is most closely related to the genera *Duma* and *Polygonum* [27,29,41].

The genus *Atraphaxis* (belonging to the Polygonaceae family) consists of over 60 species and subspecies. These species are found across a wide geographical range, extending from southeastern Europe and northeastern Africa to East Siberia, China, and Mongolia. The primary areas where the taxonomic diversity of these species is most concentrated are located in Southwest Asia and Central Asia. *Atraphaxis* was classified within the Polygonaceae family into the tribes *Polygoneae*, *Rumiceae*, *Atraphaxideae*, or *Calligoneae* based on its morphological characteristics. Nonetheless, the significant resemblance observed in the petiole and stem anatomy provides substantial evidence for classifying *Atraphaxis* within the Polygoneae tribe, alongside other genera, such as *Polygonum* L., *Polygonella michx*., *Oxygonum burch*., *Calligonum* L., and *Fagopyrum mill*. This classification strongly aligns with the findings of recent molecular studies, the most recent of which has elucidated the composition of the Polygoneae tribe within the Polygonoideae subfamily [28,29,30,41,42].

The taxonomy study of *A. laetevirens*, *A. frutescens*, *A. spinosa* L., and *A. pyrifolia* involves a comprehensive examination of these plant species within the genus Atraphaxis. Taxonomy involves the classification, identification, and relationships among organisms. *Atraphaxis* species are part of the Polygonaceae family and are often found in various geographical locations, including Central Asia and Europe.

*A. laetevirens*, known for its unique characteristics, thrives in specific environments and possesses distinctive morphological features that differentiate it from other *Atraphaxis* species. Conversely, *Atraphaxis frutescens* and *A. spinosa* L. exhibit specific growth patterns and morphological attributes that distinguish them within the genus. *A. pyrifolia*, another member of this group, has garnered attention for its diverse chemical constituents and potential pharmacological properties.

Taxonomic studies involve analyzing botanical characteristics, including leaf structures, flower morphology, stem features, and reproductive organs. Furthermore, molecular techniques, such as DNA analysis, play a pivotal role in understanding the genetic relationships among these species.

Through taxonomic investigation, researchers aim to elucidate the evolutionary history, genetic diversity, and ecological adaptations of *Atraphaxis* species. Such studies contribute significantly to botanical science, aiding in plant identification, conservation efforts, and the understanding of plant biodiversity in various ecosystems.

## 3. Distribution of Four *Atraphaxis* Species

Four *Atraphaxis* species exhibit diverse distributions across Central Asia, spanning from Afghanistan, Kazakhstan, and Kirgizstan to regions within Russia, Mongolia, parts of the Middle East, and China, showcasing their adaptability to varied geographical landscapes (Figure 2) [33]. *A. pyrifolia* spans Afghanistan, Kazakhstan, Kirgizstan, Pakistan, Tadzhikistan, Uzbekistan, and Xinjiang. *A. spinosa* L. extends across diverse territories, including Afghanistan, East European Russia, Egypt, Iran, Kazakhstan, Kirgizstan, Lebanon-Syria, Mongolia, North Caucasus, Pakistan, Palestine, Saudi Arabia, Sinai, Tadzhikistan, Transcaucasia, Turkey, Turkmenistan, Uzbekistan, West Siberia, and Xinjiang. *A. laetevirens* predominantly inhabits Altay, Kazakhstan, Krasnoyarsk, Mongolia, Tuva, Uzbekistan, and Xinjiang. Lastly, *A. frutescens* occupies regions such as Altay, North-Central China, East European Russia, Inner Mongolia, Kazakhstan, Krasnoyarsk, Mongolia, Qinghai, South European Russia, Turkmenistan, Tuva, Uzbekistan, West Siberia, and Xinjiang. These distributions highlight the wide-ranging presence of *Atraphaxis* species across Central Asia, parts of Europe, and portions of the Middle East, demonstrating their adaptability to diverse geographical and climatic conditions [43,44].

## 4. Botany of Some *Atraphaxis* Species

*Atraphaxis* species display leaves that are diverse in size and shape, reflecting their adaptability to different habitats. The leaves are typically arranged alternately along the stems. These leaves can be linear, lanceolate, or elliptical, depending on the species. One distinctive aspect of *Atraphaxis* leaves is their grayish-green coloration, which is often associated with xerophytic adaptations. The leaf surfaces may be smooth or covered with fine hairs, a characteristic known as pubescence. This pubescence can serve as a protective mechanism, reducing water loss by limiting transpiration. *Atraphaxis* species can be found in the form of compact or towering shrubs, occasionally taking the form of small subshrubs, with heights ranging from 20 cm to 300 cm. Annual shoots for *Atraphaxis* species are elongated and constricted. In the case of the majority of *Atraphaxis* species, the leaf margin typically displays a finely crenulate, undulate, flat, or slightly revolute pattern. Additionally, the ochrea in thyrses takes the form of an oblique, funnel-shaped structure, featuring a reduced leaf blade or a keel, and measuring between 2 to 7 mm in length. The position of thyrses in *Atraphaxis* species can be either terminal or lateral. The shape of the perianth in *Atraphaxis* species during the fruiting stage can be described as campanulate, with segments that are either of equal size or with inner segments significantly enlarged, tightly encasing the achene. The length of the perianth in *Atraphaxis* species typically falls within the range of 6.5 to 14.5 mm [19,23]. The segments in *Atraphaxis* species can vary in size, with some being equal, subequal, or unequal. Furthermore, the shape of these segments may be described as rotundate, reniform, broadly elliptical, broadly ovate, obtuse, flat, undulate, and occasionally, but rarely, concave. The consistence of the segments in *Atraphaxis* species is described as petaloid, indicating that they have a petal-like texture or appearance. The surface of the perianth in *Atraphaxis* species is typically smooth (glabrous), with the rare occurrence of slight papillate features at the base of the tube and segments. The edge of the segments in *Atraphaxis* species can either be papillate, meaning they have small, raised, nipple-like structures, or not papillate, having a smooth edge. Stomata on the segments of *Atraphaxis* species are either absent or very rarely found, and if present, they are typically located only at the base of the segments. The shape of the perianth tube in most species can be described as filiform, featuring a wedge-shaped or cup-shaped extension at the top. The length of the perianth tube in *Atraphaxis* species typically falls within the range of 0.5 to 7.5 mm. The length of the filiform part of the perianth tube in *Atraphaxis* species typically ranges from 0.5 to 7.0 mm [19,22]. The size of the achene in species typically measures between 2.5 to 5.5 mm in length and 1.5 to 5.0 mm in width. The shape of the achene in most species can be described as ovoid, triquetrous (having three distinct angles or faces), or lenticular (lens-shaped). The styles and stigma in *Atraphaxis* species can either be connate at the base (joined together) or free (not joined), and they may have a capitate (having a rounded, knob-like shape) or fimbriate (having fringed or filamentous structures) appearance. Moreover, the surface of the achene in species can be described as smooth, smooth-pitted (featuring small pits or depressions), minutely rugulate (with fine wrinkles or ridges), or tuberculate (having small tubercles or warty protrusions). The surface of the sporoderm in most species can be characterized as striae-perforate, with rare occurrences of being reticulato-perforate. This indicates that the surface has fine parallel lines or striations with perforations, although reticulate (net-like) perforations are occasionally present [19,20,24].

The stems of *Atraphaxis* plants can exhibit variability, with some appearing woody and others herbaceous, depending on the species and the environmental conditions. A striking feature of many *Atraphaxis* species is the presence of swollen nodes or joints along the stem. These nodes give rise to the common name “knotweed family”. These nodes serve various functions, including structural support and the storage of water and nutrients. They are often an important characteristic for identification and taxonomic classification within the genus. *Atraphaxis* species produce relatively small and often inconspicuous flowers that are typically greenish, pink, or white in color. These flowers are arranged in axillary clusters or spikes along the stems. Although the flowers may not be showy, they play a crucial role in the reproductive success of these plants. The coloration and arrangement of these flowers can vary among species, contributing to the diversity within the genus. The inflorescence structure varies among *Atraphaxis* species but is commonly a raceme or panicle. These arrangements are essential for the dispersion of flowers and the successful pollination of the plants. The diversity in inflorescence structures provides insights into the adaptation of different species to their specific environments and pollinators. The fruits of *Atraphaxis* plants are small and typically triangular to orbicular in shape. These fruits, known as achenes, each contain a single seed. One distinguishing feature of *Atraphaxis* fruits is their enclosure within the persistent calyx. This calyx, which surrounds the fruit, persists even after flowering, serving as a protective covering. It can also aid in seed dispersal, and it is an important trait for identifying species within the genus [19]. The presence or absence of pubescence is a notable characteristic in *Atraphaxis* species. Pubescence refers to the fine hairs or trichomes found on the surfaces of leaves, stems, and other plant parts. Some *Atraphaxis* species display pronounced pubescence, which contributes to their ability to reduce water loss by limiting transpiration. In contrast, others have a smoother, glabrous surface. This variability in pubescence is linked to their adaptation to specific environmental conditions [19,21].

Overall, the genus *Atraphaxis* stands as a testament to nature’s capacity for adaptation and resilience in the face of challenging environmental conditions. Its morphological characteristics, such as leaves, stems, flowers, inflorescence, fruits, growth forms, and pubescence, are essential for species identification, providing insights into their ecological roles. These plants have evolved to thrive in arid and semi-arid regions, contributing to the intricate tapestry of life in these often-harsh environments. By understanding the morphological features of *Atraphaxis*, we gain a deeper appreciation for the diversity and adaptability of this remarkable genus [19,22].

According to Table 1, *Atraphaxis* is a diverse genus within the Polygonaceae family, encompassing several species that exhibit variations in morphological characteristics. *A. laetevirens*, *A. frutescens*, *A. spinosa*, *A. pyrifolia*, *A. binaludensis*, *A. intricata*, *A. seravschanica*, and *A. radkanensis*, each display distinct morphological features, contributing to their taxonomic differentiation. *A. laetevirens* typically showcases spinescent branches with obovate to lanceolate leaves, acute leaf apices, and entire or slightly undulate leaf margins [14,25]. Conversely, *A. frutescens* presents non-spinescent branches, lanceolate to elliptic leaves with acuminate apices, and entire to serrate margins. *A. spinosa* exhibits spinescent branches with ovate to lanceolate leaves, acute apices, and entire or slightly undulate margins.

*A. pyrifolia* displays spinescent branches, obovate rhomboid leaves with rounded-obtuse apices and slightly revolute margins [14,26]. *A. binaludensis* manifests spinescent branches with obovate-rhomboid leaves having rounded-obtuse apices and slightly revolute margins. *A. intricata*, *A. seravschanica*, and *A. radkanensis* differ further in specific leaf sizes, shapes, petiole lengths, flower sizes, pedicel lengths, and fruit lengths, contributing to their distinct morphological identities within the genus [14,25,26]. These species’ morphological variations, including leaf shape, apices, margins, and branch types, contribute significantly to their taxonomic classification and ecological adaptations. Studying these morphological differences aids in better understanding species diversity, evolutionary relationships, and ecological roles within their respective habitats.

## 5. Traditional Use

Traditional medical usage of *Atraphaxis* species has a long history, especially in Asia and the Middle East (Table 2). Traditional medical systems have used these plants because of their possible medicinal benefits. *Atraphaxis* species have been utilized to treat gastrointestinal issues like diarrhea, indigestion, and stomachaches in several traditional medical systems. To treat these symptoms, the plants can be consumed as herbal decoctions or infusions. Some *Atraphaxis* species have been employed for their ability to reduce inflammation. To treat illnesses like arthritis, joint discomfort, or skin inflammations, they can be administered topically or taken internally, as herbal treatments. *Atraphaxis* species have been utilized by traditional healers in some areas to promote wound healing [34]. To speed up the healing process, the plants can be made into poultices or ointments and applied to cuts and wounds. There is some ethnopharmacological data pointing to the possibility of antidiabetic effects in *Atraphaxis* species. Traditional medicine has employed these plants’ extracts to control blood sugar levels. *Atraphaxis* species have been used to cure respiratory conditions such as asthma and coughing in several cultures. To treat respiratory ailments, infusions or extracts from these plants are consumed orally or by steam inhalation. Traditional medicine may make use of *Atraphaxis* species due to their antioxidant capabilities, which can assist the body in fighting oxidative stress [35].

## 6. Bioactive Compounds from Four *Atraphaxis* Species

Phytochemical studies of *Atraphaxis* species have revealed the presence of various bioactive compounds, although the specific phytochemical composition can vary among different species and even within the same species. Some of the common phytochemicals that have been identified in *Atraphaxis* species include polyphenols, triterpenoids, alkaloids, essential oils, phenolic acids, saponins, and lignans. *Atraphaxis* species are known to contain various polyphenolic compounds, such as flavonoids and tannins. These compounds have antioxidant properties and may contribute to the plant’s ability to protect itself from oxidative stress. Triterpenoids are a class of compounds that have been found in some *Atraphaxis* species. These compounds have various biological activities and are known for their potential anti-inflammatory and cytotoxic properties. Some *Atraphaxis* species have been reported to contain alkaloids. Alkaloids are nitrogen-containing compounds with a wide range of pharmacological activities. The presence of alkaloids in Atraphaxis species may contribute to their medicinal properties. Certain *Atraphaxis* species produce essential oils that contain volatile compounds responsible for the characteristic aroma of the plant. These essential oils may have various uses, including in traditional medicine and as flavoring agents [37]. Phenolic acids, such as caffeic acid and ferulic acid, have been detected in *Atraphaxis* species. These compounds have antioxidant and anti-inflammatory properties and are commonly found in many plant species. Saponins are glycosides found in some *Atraphaxis* species. They are known for their foaming properties and have been studied for their potential health benefits, including as anticancer agents and immune system modulators. Lignans are polyphenolic compounds with antioxidant properties. Some *Atraphaxis* species have been reported to contain lignans, which may contribute to their medicinal potential. It is important to note that the phytochemical composition of *Atraphaxis* species can vary depending on factors such as species, geographical location, and environmental conditions. Additionally, the presence of specific phytochemicals may have implications for the traditional medicinal uses of these plants in various cultures. Phytochemical studies of *Atraphaxis* species are ongoing, and researchers continue to explore the potential pharmacological properties and applications of the compounds found in these plants. These studies may lead to the development of new drugs or natural products for various purposes, including medicine, agriculture, and industry [38].

*A. frutescens* and *A. spinosa* L. are distinguished from other *Atraphaxis* species due to their rich assortment of chemical components. A study on *A. frutescens* revealed the presence of several flavonoids featuring pyrogallol B-ring structures, which are renowned for their antioxidative qualities. *A. frutescens* has yielded fisetinidol, a compound known for its potential biological activities, including its antioxidant and anti-inflammatory properties. Another compound discovered in *A. frutescens* is catechin, which is recognized for its antioxidant properties and potential health benefits. Furthermore, the aerial parts of *A. frutescens* yielded quercetin and butin, both of which are flavonoids with antioxidative and anti-inflammatory properties. Additionally, compounds such as quercetin-3-methyl ether, 5-deoxykaempferol, and β-sitosterol glucoside have been identified in *A. frutescens,* although their pharmacological properties have not been thoroughly investigated yet [33,35].

Nine compounds were obtained and identified from an ethereal extract of *A. spinosa* L. var. *sinaica*. These compounds were characterized as N-trans-p-coumaroyl-3′,4′-dihydroxyphenylethylamine, N-trans-feruloyl-3′,4′-dihydroxyphenylethylamine, (−)-fisetinidol, (−)-catechin, butin, quercetin, quercetin-3-methyl ether, 5-deoxykaempferol, and β-sitosterol glucoside. The compounds N-trans-p-coumaroyl-3′,4′-dihydroxyphenylethylamine and N-trans-feruloyl-3′,4′-dihydroxyphenylethylamine, which were isolated from natural sources for the first time, exhibited cytotoxic effects on leukemic P388 cells [34,36,37].

The comprehensive exploration of *A. pyrifolia* and *A. laetevirens* concerning their biologically active compounds remains incomplete. Nevertheless, limited research has acknowledged the presence of certain valuable compounds within these species. For instance, chrysophanol, physcion, nepodin, and emodin represent natural compounds that exist in diverse plant species, including *A. laetevirens*. Classified as anthraquinones, these compounds fall into the category of organic molecules and have displayed various pharmacological properties in scientific investigations. Specifically, chrysophanol and nepodin have gained recognition for their effectiveness in combating insects and bacteria [38].

Studies of *A. pyrifolia* have identified flavonoid glycosides, including compounds such as 7-methylgossypetin 8-β-D-glucopyranoside 3-O-α- L-rhamnopyranoside, and 7-methylgossypetin 8-β-D-glucopyranoside. However, despite their detection, extensive research into their potential pharmacological applications is yet to be conducted.

Overall, the biologically active compounds found have been discovered in select *Atraphaxis* species, showcasing the rich potential of these plants in the field of natural medicine (Table 3). These compounds, often associated with antioxidant and anti-inflammatory properties, offer promising avenues for pharmaceutical research and the development of novel drugs. The identification and isolation of these bioactive constituents hold the key to harnessing the therapeutic benefits of *Atraphaxis* species for various health applications.

Phenols, a class of organic compounds characterized by a hydroxyl (OH) group attached to an aromatic ring, exhibit diverse pharmacological activities that have significant implications for human health. These compounds are renowned for their potent antioxidant properties, effectively scavenging free radicals and reducing oxidative stress, as indicated in numerous studies [97]. Their anti-inflammatory effects make them valuable in alleviating symptoms associated with conditions such as arthritis and inflammatory bowel disease, with research supporting their role in modulating inflammatory pathways [98].

Phenolic compounds also demonstrate antimicrobial and antibacterial activity, inhibiting the growth of bacteria, fungi, and viruses. Their potential in promoting heart health through blood pressure reduction and cholesterol management is well-documented [99]. Furthermore, some phenolic compounds exhibit anti-cancer properties by inhibiting tumor cell growth and promoting apoptosis [100]. Additionally, these compounds can have neuroprotective effects and may reduce the risk of neurodegenerative diseases, such as Alzheimer’s and Parkinson’s disease [101].

Moreover, phenolic compounds play a pivotal role in supporting liver health by protecting the liver from damage and aiding in tissue regeneration. They also have anti-diabetic potential by regulating blood sugar levels and improving insulin sensitivity. The immunomodulatory effects of phenols enhance immune responses against infections and diseases, and their use in wound healing and skin health applications is based on their ability to promote tissue repair and protect the skin from oxidative damage. The versatile pharmacological activities of phenols underscore their importance in preventive and therapeutic approaches to various health conditions [102,103,104].

Flavonoids, a diverse group of polyphenolic compounds found abundantly in plants, have captivated the attention of the medical and scientific communities for their remarkable potential in medicine. These natural compounds, responsible for the vibrant colors in fruits, vegetables, and flowers, go beyond aesthetics; they hold significant therapeutic promise. Flavonoids, including quercetin, kaempferol, and catechin, are potent antioxidants. They neutralize harmful free radicals and unstable molecules that can damage cells and the DNA. This antioxidant capacity is pivotal in preventing and managing a range of diseases, including cancer, cardiovascular ailments, and neurodegenerative disorders. Inflammation is a double-edged sword, being necessary for defense but destructive in excess. Certain flavonoids like quercetin, rutin, and luteolin exhibit anti-inflammatory properties. They mitigate excessive inflammation and alleviate symptoms in conditions such as arthritis and inflammatory bowel diseases [70]. Flavonoids contribute to heart health. By reducing oxidative stress, enhancing endothelial function, and improving blood vessel elasticity, they lower the risk of hypertension, atherosclerosis, and heart disease. Prominent flavonoids such as resveratrol, found in red wine, are associated with cardiovascular benefits. Moreover, the brain benefits from flavonoids too. Compounds such as epicatechin from dark chocolate may enhance cognitive function, and quercetin could protect against neurodegenerative disorders by reducing oxidative stress and inflammation in the brain. Additionally, flavonoids have shown antiviral properties, including inhibiting the replication of viruses, such as influenza and HIV [72]. Additionally, they possess antimicrobial capabilities, which can aid in combating infections and antibiotic-resistant bacteria. Another benefit is that flavonoids have gained recognition for their potential in cancer prevention and treatment. They can induce apoptosis (programmed cell death) in cancer cells, inhibit angiogenesis (formation of blood vessels that feed tumors), and modulate signaling pathways that drive cancer growth. Certain flavonoids, such as quercetin and kaempferol, promote wound healing and skin health by aiding tissue repair, reducing inflammation, and protecting against UV-induced damage. Some flavonoids can help manage diabetes by improving insulin sensitivity and glucose metabolism, potentially reducing the risk of diabetes-related complications [73].

Flavonoids and their derivatives offer a rich palette of therapeutic possibilities, and ongoing research continues to unveil their potential applications. However, it is important to consider that the efficacy and safety of flavonoid-based treatments may vary depending on the specific compound, dosage, and individual factors. Integrating flavonoid-rich foods and supplements into a balanced diet may offer a holistic approach to harnessing their health benefits. As science explores this colorful world of natural compounds, the future of medicine may hold exciting breakthroughs thanks to flavonoids [74].

Flavonoids, specifically flavonol glycosides, such as myricitrin, have been identified in *A. frutescens* [35] and *A. spinosa* L. [37]. Myricitrin exhibits noteworthy antioxidant properties and showcases anti-inflammatory as well as anti-nociceptive effects [78]. This flavonoid compound has garnered attention for its potential role in combating oxidative stress, reducing inflammation, and mitigating pain responses within the biological system.

Another flavonoid compound, emodin 8-O-β-D-glucopyranoside, discovered in A. *frutescens* [35], displays significant pharmacological activities. Emodin 8-O-β-D-glucopyranoside has shown promise as an anticancer agent [79], demonstrating potential in the fight against cancerous cell proliferation. Additionally, this compound exhibits neuroprotective properties [80], indicating its potential role in preserving neural health and potentially mitigating neurodegenerative conditions.

These flavonoid glycosides, found within *Atraphaxis* species, particularly myricitrin and emodin 8-O-β-D-glucopyranoside in *A. frutescens*, showcase a range of pharmacological effects. Their antioxidant, anti-inflammatory, anti-nociceptive, anticancer, and neuroprotective attributes demonstrate their potential therapeutic significance in various medical applications and merit further exploration in the domain of pharmacology and medical research. However, other flavonoid glycosides, specifically 3,4,5-trimethoxyphenyl 1-O-β-D-glucopyranoside, 8-O-β-D-glucopyranosyl-7-O-methyl-3-O-α-L-rhamnopyranosylgossypetin, 8-O-acetyl-7-O-methyl-3-O-α-L-rhamnopyranosylgossypetin, 7-O-methyl-3-O-α-rhamnopyranosylgossypetin, quercitrin, quercetin 3-O-β-D-glucuronide-6-O-methyl ester, quercetin-3-O-β-D-glucuronide, afzelin, 7-methylgossypetin 8-β-D-glucopyranoside 3-O-α-L-rhamnopyranoside, and 7-methylgossypetin 8-β-D-glucopyranoside, have not been thoroughly studied for their pharmacological properties, and their effects remain largely unclear.

## 7. Pharmacological Activities of Found Compounds from Four *Atraphaxis* Species

A vast reservoir of selected species remains untapped in terms of phytochemical constituents, as well as pharmacology, and this is the research gap for future studies. Further investigations of *Atraphaxis* species, along with their phytochemical constituents, are necessary to completely understand the molecular mechanisms of their action in vivo and in vitro and to ensure the plant extracts are safe for human use. According to findings related to bioactive compounds, it is possible to use *Atraphaxis* species for different purposes, as shown in Figure 3.

*Atraphaxis* species are known to contain polyphenols, such as flavonoids and tannins, which are well-regarded for their potent antioxidant properties. These compounds have the ability to counteract oxidative stress, inflammation, and cellular damage, making them valuable assets in the realm of health and medicine. Their therapeutic potential extends to conditions associated with these processes, including cardiovascular diseases and cancer. Another class of compounds found in *Atraphaxis* species includes triterpenoids. These compounds exhibit anti-inflammatory and antioxidant attributes, which can be invaluable for addressing inflammatory disorders and promoting overall health and wellness. The anti-inflammatory properties make them particularly attractive for conditions involving excessive inflammation, such as arthritis [105,106]. In addition to polyphenols and triterpenoids, *Atraphaxis* species may also contain alkaloids with diverse pharmacological actions. Alkaloids are nitrogen-containing compounds with a wide range of potential effects, including analgesic and antimicrobial properties. These compounds have shown promise in managing pain and combating various pathogens. Moreover, *Atraphaxis* species may harbor lignans with potential hormonal activity. Lignans are phytochemicals with estrogenic effects and may hold promise in addressing hormonal imbalances and managing menopausal symptoms [33,34,35,36,37].

In summary, *Atraphaxis* species represent a rich source of bioactive molecules with diverse pharmacological activities. The compounds found in these plants have the potential to contribute to the development of novel medicines and health-promoting products [107,108]. However, it is essential to note that the specific pharmacological activities and applications may vary among different *Atraphaxis* species and may require further research and investigation. Continued scientific exploration of these natural compounds may unlock their full therapeutic potential, leading to valuable contributions to healthcare and wellness. The pharmacological activities of the compounds found in *Atraphaxis* species, namely *A. laetevirens*, *A. frutescens*, *A. spinosa L*., and *A. pyrifolia*, encompass a diverse range of potential effects, although comprehensive studies on these activities are still needed for a more definitive understanding. Some identified compounds have shown promising pharmacological properties in preliminary research. For instance, known to be present in A. frutescens, myricitrin exhibits antioxidant, anti-inflammatory, and anti-nociceptive effects [35]. Emodin 8-O-β-D-glucopyranoside was discovered in *A. frutescens*. This compound demonstrates potential as both an anticancer and a neuroprotective agent [35,79,80]. Compounds such as 3,4,5-Trimethoxyphenyl 1-O-β-D-glucopyranoside, 8-O-β-D-glucopyranosyl-7-O-methyl-3-O-α-L-rhamnopyranosylgossypetin, 8-O-acetyl-7-O-methyl-3-O-α-L-rhamnopyranosylgossypetin, 7-O-methyl-3-O-α-rhamnopyranosylgossypetin, quercitrin, quercetin 3-O-b-D-glucuronide-6″-methyl ester, quercetin-3-O-bD-glucuronide, afzelin, 7-methylgossypetin 8-β-D-glucopyranoside 3-O-α-L-rhamnopyranoside, and 7-methylgossypetin 8-β-D-glucopyranoside are found in *Atraphaxis* species, but their pharmacological activities have not been fully elucidated [38].

The identified compounds display potential pharmacological activities, ranging from antioxidant and anti-inflammatory effects to anticancer and neuroprotective properties. However, further in-depth investigations and studies are necessary to fully comprehend and establish the therapeutic potential and mechanisms of action of these compounds from Atraphaxis species.

The pharmacological activities of *Atraphaxis* species may vary depending on the specific species and the compounds they contain [109]. Commonly used solvents for extracting bioactive compounds from *Atraphaxis* species include ethanol, methanol, water, and sometimes a combination of these solvents. Techniques such as Soxhlet extraction and liquid-liquid extraction have been employed to extract bioactive compounds from *Atraphaxis* plants [35,37,38,104]. As a result of studying various technologies for obtaining extracts from *Atraphaxis* species, various biologically active compounds have been discovered, and their pharmacological activities are given in Table 4.

## 8. Materials and Methods

All ethnobotanical, phytochemical, pharmacological, and clinical data were collected from online journals, magazines, and books (all of which were published in English, Arabic, and Persian) from 1980 to 2023. Electronic databases such as Google, Google Scholar, PubMed, Science Direct, Researchgate, and other online collections were used. *Atraphaxis*, Polygonaceae, *Atraphaxis laetevirens*, *Atraphaxis frutescens*, *Atraphaxis spinosa* L., *Atraphaxis pyrifolia*, *Atraphaxis binaludensis*, *Atraphaxis intricata*, *Atraphaxis seravschanica*, and *Atraphaxis radkanensis* were applied as keywords on the above-mentioned engines, and their use in pharmacology and ethnomedicine were described in this manuscript.

## 9. Conclusions

In the world of natural compounds and traditional medicine, *Atraphaxis* species represent a fascinating and largely untapped resource with the potential for significant pharmacological applications. The compounds found within these plants, including polyphenols, triterpenoids, alkaloids, essential oils, and lignans, possess a diverse range of pharmacological activities that make them valuable in various healthcare and wellness contexts. However, despite the promising characteristics of these compounds, our understanding of their full potential remains incomplete. A vast reservoir of selected species within the *Atraphaxis* genus remains untapped in terms of their phytochemical constituents and the pharmacological effects they may offer. This presents a significant research gap that must be addressed in future studies. To unlock the true potential of *Atraphaxis* species and their bioactive compounds, further investigations are necessary. Researchers need to investigate the molecular mechanisms of these compounds in both in vivo and in vitro settings. Understanding how these compounds interact with biological systems is crucial in harnessing their therapeutic benefits fully. Such investigations can shed light on how these natural compounds can be utilized to treat and manage various health conditions, from cardiovascular diseases to inflammatory disorders, and even cancer. Moreover, ensuring the safety of plant extracts derived from *Atraphaxis* species is of paramount importance. Before these compounds can be integrated into human healthcare, comprehensive toxicological studies and clinical trials are essential. The safety and efficacy of these compounds must be rigorously evaluated to avoid any potential adverse effects and to provide peace of mind to healthcare practitioners and patients alike. In essence, the research journey with *Atraphaxis* species is multifaceted. It begins with unlocking the mysteries of the phytochemical constituents that make these plants unique and valuable. With this knowledge in hand, researchers can then investigate the molecular mechanisms of action, both within living organisms (in vivo) and in controlled laboratory settings (in vitro). This comprehensive approach is vital for establishing the safety and efficacy of these natural compounds and for harnessing their therapeutic potential. In conclusion, *Atraphaxis* species have emerged as a promising resource in the field of natural compounds and traditional medicine. Yet, the road ahead is filled with both exciting opportunities and challenges. The untapped reservoir of species and the need for comprehensive research present a compelling case for future studies. By exploring the full range of phytochemical constituents and understanding their pharmacological properties, researchers can pave the way for a new era of healthcare and wellness that leverages the untapped potential of these remarkable plants. However, caution, diligence, and rigorous research are essential to ensure that compounds derived from *Atraphaxis* species are safe, effective, and ready for integration into healthcare practices. Studying *A. laetevirens*, *A. frutescens*, *A. spinosa L.,* and *A. pyrifolia* is crucial, as it provides valuable insights into the pharmacological properties of their bioactive compounds, potentially offering novel avenues for therapeutic applications and contributing to the development of new drugs or natural remedies in medicine and healthcare.

## Figures and Tables

**Figure 1 molecules-29-00910-f001:**
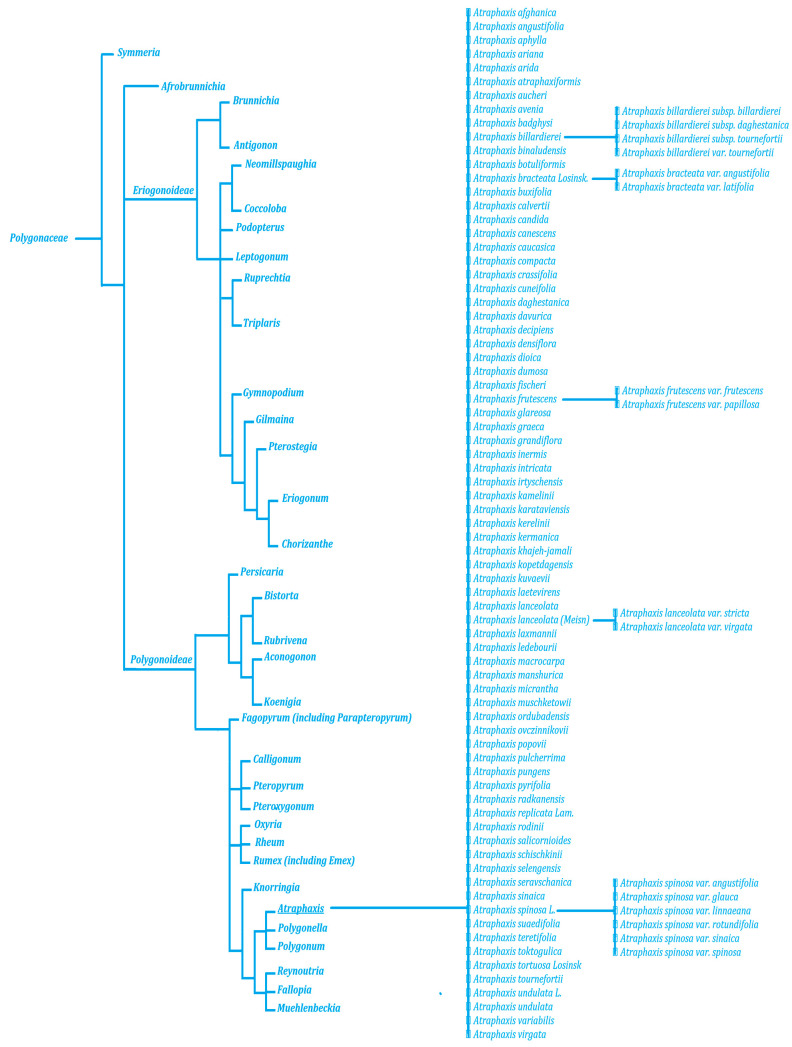
The phylogenic tree of *Atraphaxis* species made based on the literature survey [17,18,19,20,27,29,40,41].

**Figure 2 molecules-29-00910-f002:**
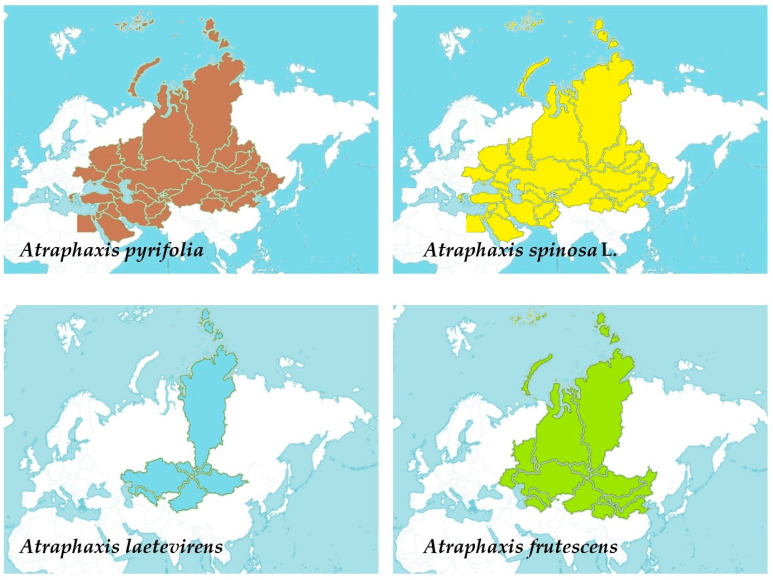
The distribution of four different *Atraphaxis* species using shaded colors: *Atraphaxis pyrifolia* represented in brown, *Atraphaxis spinosa* L. in yellow, *Atraphaxis laetevirens* in light blue, and *Atraphaxis frutescens* in green.

**Figure 3 molecules-29-00910-f003:**
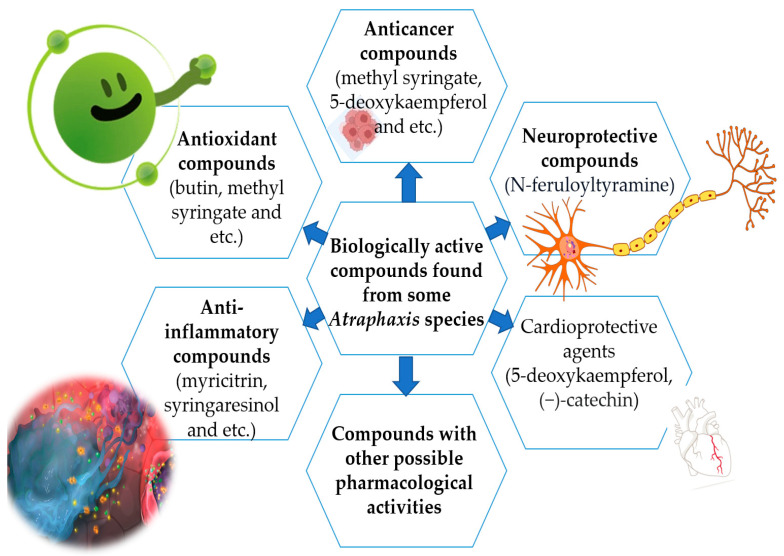
Biological active compounds with pharmacological activities.

**Table 1 molecules-29-00910-t001:** Diagnostic morphological characters and distribution area of well-studied *Atraphaxis* species.

Name of *Atraphaxis* Species	Morphological Characters	Distribution
*A. laetevirens*	Branches: spinescent; Leaf size (mm): 6–8 × 4–5; Leaf shape: obovate to lanceolate; Leaf apex: acute; Leaf margin: entire or slightly undulate; Petiole (mm): 0.5–1.5; Flower size (mm): 5–7 × 4–5; Pedicel length (mm): 2–4; Fruit length (mm): 3–5 [14,25]	Afghanistan, Kazakhstan, Kirgizstan, Pakistan, Tadzhikistan, Uzbekistan, Xinjiang [33]
*A. frutescens*	Branches: non-spinescent; Leaf size (mm): 8–10 × 5–7; Leaf shape: lanceolate to elliptic; Leaf apex: acuminate; Leaf margin: entire to serrate; Petiole (mm): 1–2; Flower size (mm): 7–9 × 5–7; Pedicel length (mm): 3–5; Fruit length (mm): 4–6 [14,25]	Afghanistan, Kazakhstan, Kirgizstan, Pakistan, Tadzhikistan, Uzbekistan, Xinjiang [33]
*A. spinosa* L.	Branches: spinescent; Leaf size (mm): 4–6 × 3–4; Leaf shape: ovate to lanceolate; Leaf apex: acute; Leaf margin: entire or slightly undulate; Petiole (mm): 0.3–0.8; Flower size (mm): 5–7 × 4–5; Pedicel length (mm): 1.5–2.5; Fruit length (mm): 2–3 [14,25]	Afghanistan, Kazakhstan, Kirgizstan, Pakistan, Tadzhikistan, Uzbekistan, Xinjiang [33]
*A. pyrifolia*	Branche: spinescent; Leaf size (mm): 15–25 × 210–213; Leaf shape: spathulate-broadly obovate; Leaf apex: rounded-obtuse; Leaf margin: slightly revolute; Petiole (mm): 3–5; Flower size (mm): 6–7 × 7–8; Pedicel length (mm): 2.5–4; Fruit length (mm): ±3 [14,26]	Afghanistan, Kazakhstan, Kirgizstan, Pakistan, Tadzhikistan, Uzbekistan, Xinjiang [33]

**Table 2 molecules-29-00910-t002:** Traditional usage of *Atraphaxis* species in ethnomedicine.

Name of *Atraphaxis* Species Used in Ethnomedicine	Health Disorders and Diseases Treated in Ethnomedicine with *Atraphaxis* Species	Countries
*A. binaludensis*	to reduce inflammation, such as arthritis, joint discomfort, or skin inflammation	Iran, Kazakhstan, Uzbekistan [33,35]
*A. laetevirens*	to alleviate gastrointestinal problems such as stomachaches and digestive disturbances	Iran, Kazakhstan, Uzbekistan [33,35]
*A. frutescens*	to alleviate gastrointestinal problems such as stomachaches and digestive disturbances	Iran, Kazakhstan, Uzbekistan [33,35]
*A. spinosa* L.	to alleviate gastrointestinal issues, including stomach discomfort, digestive disturbances, and ailments related to intestinal health	Algeria, Kazakhstan, Uzbekistan [33,35]
*A. intricata*	to alleviate arthritis, joint discomfort, or skin inflammation	Iran [33,34]
*A. seravschanica*	wound healing, skin inflammation	Kazakhstan, Tadzhikistan, Turkmenistan, Uzbekistan [17,33,35]
*A. pyrifolia*	to cure respiratory conditions such as asthma and coughing, to cure various wounds and skin inflammation	Afghanistan, Kazakhstan, Kirgizstan, Pakistan, Tadzhikistan, Uzbekistan, China [33,35]
*A. radkanensis*	To treat respiratory ailments and infusions, and to cure respiratory conditions such as asthma and coughing	Iran [33,34]

**Table 3 molecules-29-00910-t003:** Biologically active compounds found in some *Atraphaxis* species.

No.	Name	Structures	Studied *Atraphaxis* Specie(s)	Chemical Classes	Known Pharmacological Activities
1	dihydroconiferyl alcohol	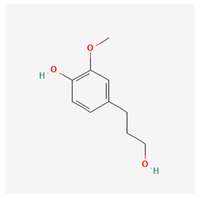	*A. spinosa* L. [34,36,37]	Phenols (phenolic alcohols)	Antioxidant [45] and anti-inflammatory [46]
2	methyl syringate	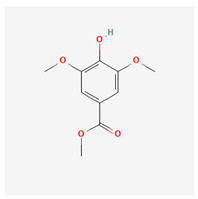	*A. frutescens* [35]	Phenols (phenolic esters)	Antioxidant [47], anti-inflammatory [48], and anticancer [49]
3	N-feruloyl dopamine, trans-	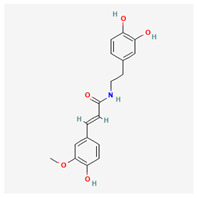	*A. frutescens* [35], *A. spinosa* L. [37]	Phenols (phenolic amides)	–
4	N-feruloyltyramine	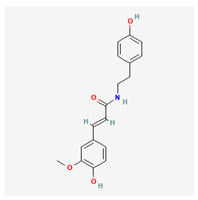	*A. frutescens* [35], *A. spinosa* L. [37]	Phenols (phenolic amides)	Antioxidant [50], neuroprotective agent [51], and anti-inflammatory [52]
5	N-p-trans-coumaroyltyramine	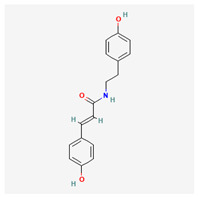	*A. frutescens* [35]	Phenols (phenolic amides)	–
6	chrysophanol	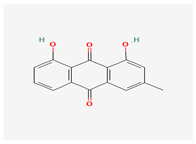	*A. laetevirens* [38]	Polyphenols (anthraquinones)	Antibacterial [38]
7	physcion	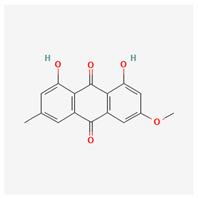	*A. laetevirens* [38]	Polyphenols (anthraquinones)	–
8	nepodin	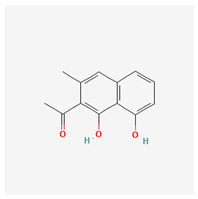	*A. laetevirens* [38]	Polyphenols (anthraquinones)	Antibacterial [38]
9	emodin	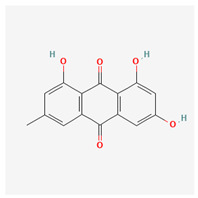	*A. laetevirens* [38]	Polyphenols (anthraquinones)	–
10	quercetin-3-methyl ether	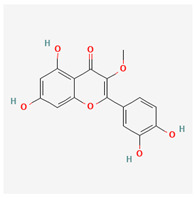	*A. frutescens* [35]	Flavonoids (flavonols)	–
11	butin	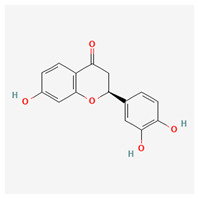	*A. spinosa* L. [36]	Flavonoids (flavonols)	Antioxidant [53], neuroprotective agent [54], anti-inflammatory [55], and anticancer [56], hepatoprotective agent [57]
12	5-deoxykaempferol	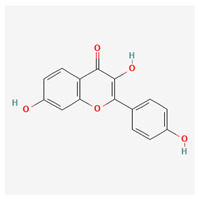	*A. spinosa* L. [34,36,37]	Flavonoids (flavonols)	Antioxidant [58], anti-inflammatory [58], anticancer [59], neuroprotective agent [60], and cardioprotective agent [61]
13	8-acetoxy-3,3′,4′,5,5′-pentahydroxy-7-methoxyflavone	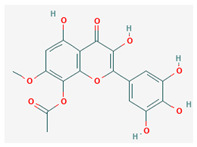	*A. spinosa* L. [34,36,37]	Flavonoids (flavonols)	–
14	3,3′,4′,5,5′,7,8-heptahydroxyflavone	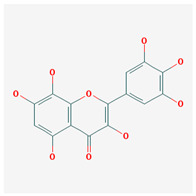	*A. spinosa* L. [34,36,37]	Flavonoids (flavonols)	–
15	(−)-fisetinidol	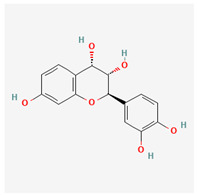	*A. frutescens* [35], *A. spinosa* L. [36,37]	Flavonoids (flavonols)	Hypoglycemic agent and hepatoprotective agent [62], antiparasitic and anticancer [63], antioxidant [64]
16	afzelechin	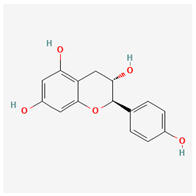	*A. frutescens* [35]	Flavonoids (flavonols)	Antioxidant [65,66]
17	(+)-dihydrokaempferol	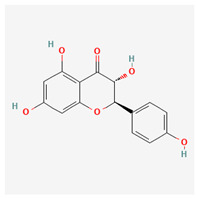	*A. frutescens* [35]	Flavonoids (flavonols)	–
18	aromadendrin	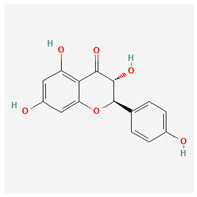	*A. frutescens* [35]	Flavonoids (flavonols)	–
19	epigallocatechin	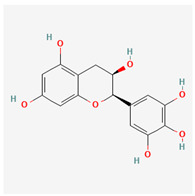	*A. frutescens* [35]	Flavonoids (flavonols)	–
20	(−)-epigallocatechin gallate	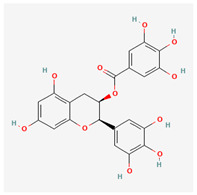	*A. frutescens* [35]	Flavonoids (catechins)	Antioxidant [67], anti-inflammatory [68], anticancer [69], neuroprotective agent [70], and anti-diabetic agent [71]
21	(−)-epicatechin	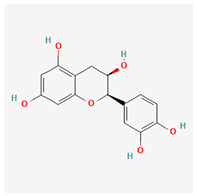	*A. frutescens* [35]	Flavonoids (catechins)	Neuroprotective agent and cardioprotective agent [72], anti-inflammatory [73]
22	gallocatechin	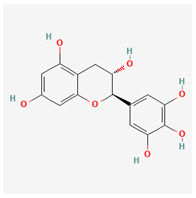	*A. frutescens* [35]	Flavonoids (catechins)	Antioxidant [67], anti-inflammatory [68], anticancer [69], neuroprotective agent [70], and anti-diabetic agent [71]
23	(−)-catechin	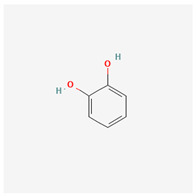	*A. frutescens* [35]	Flavonoids (catechins)	Antioxidant and anti-inflammatory [74], anticancer [75], neuroprotective agent [76] and cardioprotective agent [77]
24	myricitrin	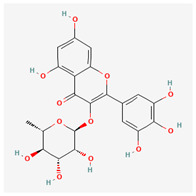	*A. frutescens* [35], *A. spinosa* L. [37]	Flavonoids (flavonol glycosides)	Antioxidant, anti-inflammatory, anti-nociceptive [78]
25	emodin 8-O-β-D-glucopyranoside	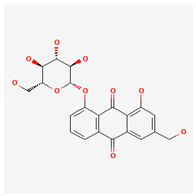	*A. frutescens* [35]	Flavonoids (flavonol glycosides)	Anticancer [79], neuroprotective agent [80,81]
26	quercitrin	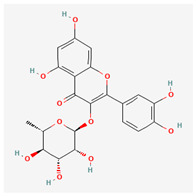	*A. spinosa* L. [37]	Flavonoids (flavonol glycosides)	–
27	afzelin	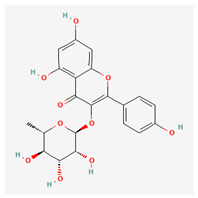	*A. spinosa* L. [37]	Flavonoids (flavonol glycosides)	–
28	syringaresinol	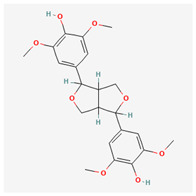	*A. frutescens* [35]	Lignans	Anti-inflammatory [82], antioxidant [83], and anticancer [84]
29	nikoenoside	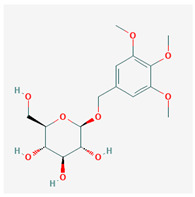	*A. frutescens* [35]	Lignans	–
30	dehydroconiferyl alcohol	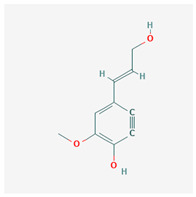	*A. frutescens* [35]	Lignans	Anti-inflammatory [82]
31	β-sitosterol glucoside	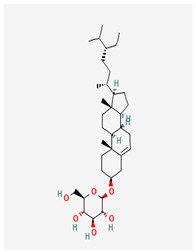	*A. frutescens* [35]	Stigmastenes	Anticancer [85], analgesic [86], apoptogenic [87], ameliorates insulin resistance and oxidative stress [88], anti-inflammatory [89], sustains normal function of the immune system [88]
32	lucidulactone A	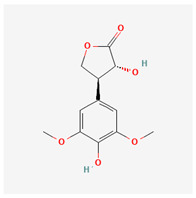	*A. spinosa* L. [37]	Terpenoids (sesquiterpenoid)	–
33	methyl gallate	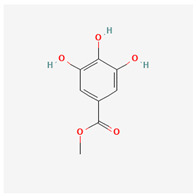	*A. spinosa* L. [37]	Gallates	anti-tumor, anti-inflammatory, anti-oxidant, neuroprotective, hepatoprotective, and anti-microbial activities [90]
34	gallic acid	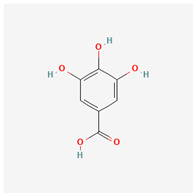	*A. spinosa* L. [37]	Gallates	anticancer [91], anti-inflammatory, and antioxidant [92]
35	a-linolenic acid	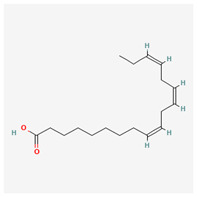	*A. spinosa* L. [37]	Fatty acids	anti-inflammatory [93]; a precursor to EPA and DHA, ALA contributes to brain health [94]
36	loliolide	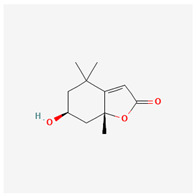	*A. spinosa* L. [37]	Lactones (cyclic lactones)	neuroprotective agent and anti-inflammatory [95], antioxidant [96]

The structural depictions of substances have been taken from the PubChem website (https://pubchem.ncbi.nlm.nih.gov accessed on 10 January 2024).

**Table 4 molecules-29-00910-t004:** The extraction parameters for four *Atraphaxis* species and their pharmacological activities.

Technologies for Obtaining	Extraction Parameters	Methods for Determining Biologically Active Substances	Pharmacological Activity	*Atraphaxis* Species	Reference
Soaking method	extracted with CH_2_Cl_2_	HPLC	the extract has antimicrobial activity against *Streptococcus iniae* and the identified compound chrysophanol had strong activity against *Flavobacterium columnare*	*A. laetevirens*	[38]
Liquid-liquid extraction	extracted with ethanol	NMR spectroscopy, HPLC	1. antioxidant activities 2. insect phenoloxidase inhibition activity, tyrosinase inhibition activity	*A. frutescens*	[35]
Liquid-liquid extraction	extracted with methanol	NMR spectroscopy	–	*A. spinosa* L.	[37]
Liquid-liquid extraction	extracted with methanol	NMR spectroscopy, UV spectroscopy	–	*A. pyrifolia*	[104]

## Data Availability

Not applicable.

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
