# Peer review of "Exploring Four *Atraphaxis* Species: Traditional Medicinal Uses, Phytochemistry, and Pharmacological Activities"

_molecules, 2024, doi:10.3390/molecules29040910_

Round 1

Reviewer 1 Report (Previous Reviewer 1)

Comments and Suggestions for Authors

Considering the title of the manuscript, the content of sections 3, 4 and 5 is too general and extensive, it should be reduced considering the information concerning the four species included, only. Resolve comments found in the manuscript.

Author Response

  1. comment

Considering the title of the manuscript, the content of sections 3, 4 and 5 is too general and extensive, it should be reduced considering the information concerning the four species included, only.

Response: Sections 3, 4 and 5 have been revised and some sort of information (about other species of Atraphaxis) were excluded.

  1. Comment

Resolve comments found in the manuscript.

Response: All the comments that were found were fixed as per the suggestions.

Reviewer 2 Report (New Reviewer)

Comments and Suggestions for Authors

This manuscript presents a comprehensive review of Atraphaxis species and their bioactive applications. While the content is thorough, I recommend the following major revisions before considering acceptance:

  1. 1. The manuscript delves extensively into the family tree, distribution, and botany of Atraphaxis species, which might be beyond the scope of the Molecules journal. To better align with the journal's focus, the authors should consider summarizing and condensing these details, placing greater emphasis on the chemical composition and bioactivity of Atraphaxis.

  2. 2. Regarding Figure 4, it appears unnecessary for the context. I propose integrating the compound structures from Figure 4 into Table 3 and subsequently omitting the figure. This adjustment would streamline the reading experience, making it more direct and reader-friendly.

Author Response

Comments and Suggestions for Authors

This manuscript presents a comprehensive review of Atraphaxis species and their bioactive applications. While the content is thorough, I recommend the following major revisions before considering acceptance:

  1. The manuscript delves extensively into the family tree, distribution, and botany of Atraphaxis species, which might be beyond the scope of the Molecules journal. To better align with the journal's focus, the authors should consider summarizing and condensing these details, placing greater emphasis on the chemical composition and bioactivity of Atraphaxis.

Response: All sections have been revised and extensive information from sections such as taxonomic insights, distribution, and botany were excluded. The section about biologically active compounds was also revised and table 3 has been improved with additions of chemical structures.

  1. Regarding Figure 4, it appears unnecessary for the context. I propose integrating the compound structures from Figure 4 into Table 3 and subsequently omitting the figure. This adjustment would streamline the reading experience, making it more direct and reader-friendly.

The section about biologically active compounds was also revised and table 3 has been improved with additions of chemical structures.

Round 2

Reviewer 1 Report (Previous Reviewer 1)

Comments and Suggestions for Authors

With the change of title and the modifications made, the manuscript turned out much better.

Reviewer 2 Report (New Reviewer)

Comments and Suggestions for Authors

Manuscript has been improved and I don't have further comments

This manuscript is a resubmission of an earlier submission. The following is a list of the peer review reports and author responses from that submission.

Round 1

Reviewer 1 Report

Comments and Suggestions for Authors

The taxonomic information, as well as the ecological and distribution information of the genus, should be concise, only to provide context and to discuss the distribution patterns of metabolites based on the history of the genus or the environmental factors where they develop, in relation to its distribution. Due to the title of the manuscript, the “Botany” section should be removed.

It is important to include tables on the information found and analyzed in the manuscript, by species, regarding the traditional use, phytochemistry, and pharmacology of the genus, which give an idea of ​​the effort made to find the information proposed in the method.

From the text, it appears that only five of the 60 species have been studied, which implies that the tables would not be extensive.

For example, the traditional table could be included for each species used, the regions where they are used, the diseases in which they are used, the part used, the used method,  and the references.

In Fig. 1. Include the reference from where the phylogenetic tree was taken.

The foot of Fig. 2 is very long, in any case, the information should be in the main text.

Author Response

Response to Reviewer 1 Comments

Comments 1: The taxonomic information, as well as the ecological and distribution information of the genus, should be concise, only to provide context and to discuss the distribution patterns of metabolites based on the history of the genus or the environmental factors where they develop, in relation to its distribution. Due to the title of the manuscript, the “Botany” section should be removed.

Response 1: The taxonomic information, distribution, and botany have been shortened. The “Botany” section has been mentioned in the title.

Comments 2: It is important to include tables on the information found and analyzed in the manuscript, by species, regarding the traditional use, phytochemistry, and pharmacology of the genus, which give an idea of ​​the effort made to find the information proposed in the method.

Response 2: All necessary information has been improved as suggested by the reviewer.

Comments 3: From the text, it appears that only five of the 60 species have been studied, which implies that the tables would not be extensive

Response 3: Yes, a few species have been studied in terms of phytochemistry and their traditional use in different countries remains unclear. That is why in the text we tried to provide all possible studied plant species of Atraphaxis. This work is going to be a potential contribution.

Comments 4: For example, the traditional table could be included for each species used, the regions where they are used, the diseases in which they are used, the part used, the used method,  and the references.

Response 4: The traditional use table has been added (Table 2)

Comments 5: In Fig. 1. Include the reference from where the phylogenetic tree was taken.

Response 5: The references have been included (Please see Figure 1).

Comments 6: The foot of Fig. 2 is very long, in any case, the information should be in the main text.

Response 6: It has been fixed (Please see Figure 2).

Reviewer 2 Report

Comments and Suggestions for Authors

Are extracts from this plant an ingredient of a drug approved for human treatment?

Are extracts from this plant an ingredient of a drug approved for human treatment?

Please describe examples of the use of this plant or extracts based on it in the process of food fortification. Are there such examples in the literature?

Author Response

Response to Reviewer 2 Comments

Comments 1: Are extracts from this plant an ingredient of a drug approved for human treatment?

Response 1: The realm of medicinal plants has long been a source of fascination for scientists and medical practitioners alike. Atraphaxis, a genus of flowering plants known for its various species, has been under scrutiny for its potential medicinal properties. Extracts derived from these plants have shown promising capabilities in addressing numerous health complications, holding the prospect of breakthroughs in medical treatment. However, despite their evident therapeutic potential, these extracts have yet to be officially sanctioned for human treatment.

Comments 2: Are extracts from this plant an ingredient of a drug approved for human treatment?

Response 2: The realm of medicinal plants has long been a source of fascination for scientists and medical practitioners alike. Atraphaxis, a genus of flowering plants known for its various species, has been under scrutiny for its potential medicinal properties. Extracts derived from these plants have shown promising capabilities in addressing numerous health complications, holding the prospect of breakthroughs in medical treatment. However, despite their evident therapeutic potential, these extracts have yet to be officially sanctioned for human treatment.

Comments 3: Are there such examples in the literature?

Response 3: A literature survey did not yield precise information about any specific extract that had been approved as a drug for human treatment.

Reviewer 3 Report

Comments and Suggestions for Authors

The manuscript aims to review different aspects of the genus Atraphaxis. In my opinion it could be an interesting contribution for later researchers. Therefore, I congratulate the authors.

I list below some issues that we consider should be improved by the authors so that the manuscript can be published in Molecules:

The tables and figures presented by the authors seem colorful and interesting to me, making reading much more pleasant. However, the image of the species A. pyrifolia (figure 3) is not good and should be changed.

Section number 6, Traditional use, seems very brief to me. Consider that it should be expanded much further by discussing the quotes that appear in that section. You can also include the text that appears in the Introduction related to traditional knowledge. Some type of table could be included showing the medicinal uses collected to date for the different species. This work that I propose can serve to enrich or validate biological activities of phytochemical studies already carried out.

The scientific names of the species must appear in full, including the authors, the first time they are cited in the text.

This review seems correct to me and may be the beginning of more in-depth studies on each of the species of the genus Atraphaxis.

Author Response

Response to Reviewer 3 Comments

Comments 1: The tables and figures presented by the authors seem colorful and interesting to me, making reading much more pleasant. However, the image of the species A. pyrifolia (figure 3) is not good and should be changed.

Response 1: Figure 3a has been replaced (Please see Figure 3a)

Comments 2: Section number 6, Traditional use, seems very brief to me. Consider that it should be expanded much further by discussing the quotes that appear in that section. You can also include the text that appears in the Introduction related to traditional knowledge. Some type of table could be included showing the medicinal uses collected to date for the different species. This work that I propose can serve to enrich or validate biological activities of phytochemical studies already carried out.

Response 2: Table 2 has been added to section 6, in which it is possible to find information about most known Atraphaxis species used in various cultures.

Comments 3: The scientific names of the species must appear in full, including the authors, the first time they are cited in the text.

Response 3: The response about the scientific names of species and authors has been revised and corrected as suggested.

Comments 4: This review seems correct to me and maybe the beginning of more in-depth studies on each of the species of the genus Atraphaxis.

Response 4: We really appreciate and hope that this work can support our current and future investigations into Atraphaxis species.

Round 2

Reviewer 1 Report

Comments and Suggestions for Authors

The title says: distribution and botany of the genus Atraphaxis, in this sense the manuscript falls short since the genus has 60 species and the only distribution map only illustrates the one of A. pyrifolia; a map of the distribution of the genus should be presented and, consequently, discuss this topic. Regarding the information included in the botany section, it also suffers from a lack of information, taking into account the total number of species. In this sense, a key that includes all 60 species is needed, since it is the most summarized way to include the morphological information of a taxon, highlighting the differences between them. In distribution and taxonomy (botany) works, it is necessary to show results, and discuss them in such a way that the morphological and biogeographic patterns of a taxon of any category are shown; in this case: a genus. Both topics can be, in themselves, a separate work.

On the other hand, the distribution patterns of secondary metabolites and their pharmacological activity by species are not shown. The same happens in the information on traditional use, it is important to highlight, by species, in which community it is used, what part they use, how they prepare it, and for what conditions it is effective.

Reviewer 3 Report

Comments and Suggestions for Authors

The authors have reviewed and corrected the manuscript, following my instructions and considering that the manuscript has improved.

Consider that section 6, Traditional Uses, already meets the minimum requirements. However, I encourage the authors to delve deeper into traditional knowledge about the genus Atraphaxis. Several works can be published in the field of Ethnobotany. Furthermore, the results of these possible studies can give rise to future bioproducts.